# Sodium azide mutagenesis induces a unique pattern of mutations

Chaochih Liu[1], Giulia Frascarelli[1], Adrian O. Stec[1], Shane Heinen[1], Li Lei[2], Skylar R. Wyant[3], Erik Legg[4], Monika Spiller[5], Gary J. Muehlbauer[1], Kevin P. Smith[1], Justin C. Fay[6], Peter L. Morrell[1]*

1 Department of Agronomy and Plant Genetics, University of Minnesota, St. Paul, Minnesota, United States of America, 2 United States of America Department of Energy Joint Genome Institute, Lawrence Berkeley National Laboratory, Berkeley, California, United States of America, 3 Department of Ecology & Evolutionary Biology, University of California, Irvine, California, United States of America, 4 Syngenta Crop Protection Inc., Greensboro, North Carolina, United States of America, 5 KWS LOCHOW GmbH, Northeim, Germany, 6 Department of Biology, University of Rochester, Rochester, New York, United States of America

* pmorrell@umn.edu

## Abstract

The nature and effect of mutations are of fundamental importance to the evolutionary process. The generation of mutations with mutagens has also played important roles in genetics. Applications of mutagens include dissecting the genetic basis of trait variation, inducing desirable traits in crops, and understanding the nature of genetic load. Previous studies of sodium azide-induced mutations have reported single nucleotide variants (SNVs) found in individual genes. To characterize the nature of mutations induced by sodium azide, we analyze whole-genome sequencing (WGS) of 11 barley lines derived from sodium azide mutagenesis, where all lines were selected for diminution of plant fitness owing to induced mutations. We contrast observed mutagen-induced variants with those found in standing variation in WGS of 13 barley landraces. Here, we report indels that are two orders of magnitude more abundant than expected based on nominal mutation rates. We found induced SNVs are very specific, with C→T changes occurring in a context followed by another C on the same strand (or the reverse complement). The codons most affected by the mutagen include the sodium azide-specific CC motif (or the reverse complement), resulting in a handful of amino acid changes and few stop codons. The specific nature of induced mutations suggests that mutagens could be chosen based on experimental goals. Sodium azide would not be ideal for gene knockouts but will create many missense mutations with more subtle effects on protein function.

**Data availability statement:** SRA numbers: WGS Morex and Morex treated with sodium azide BioProject PRJNA849997. WGS barley landraces BioProject PRJNA674330. ONT of Morex and Morex treated with sodium azide BioProject PRJNA967725. Github: https://github.com/MorrellLAB/Barley_Mutated DRUM: https://doi.org/10.13020/sewd-qq35.

**Funding:** This study was supported by a University of Minnesota Informatics Institute MnDRIVE Graduate Assistantship award to Chaochih Liu, the National Science Foundation (IOS-1339393 to PLM, JCF, and KPS), US Department of Agriculture Biotechnology Risk Assessment Research Grants Program (USDA BRAG 2023-33522-41008 to PLM), and the Minnesota Agricultural Experiment Station fund (MIN-13-122 to PLM). Syngenta Crop Protection Inc. covered the sequencing costs for a portion of our sample. The funders had no role in study design, data collection and analysis, decision to publish, or manuscript preparation.

**Competing interests:** The authors have declared that no competing interests exist.

## Author summary

Sodium azide is frequently used as a mutagen for experimental studies in plants. It induces primarily C→T changes. We find that these most often occur when a cytosine (C) is followed by another cytosine. In coding sequence, this results in a limited yet distinct set of amino acid alterations that differ from natural variants in barley. Notably, harmful mutations prevalent in sodium azide-treated samples include glycine to aspartic acid and proline to serine, while untreated landraces exhibit distinct sets of putatively harmful changes. The mutated lines demonstrate an average yield reduction of 37.7% due to induced single nucleotide variants (SNVs) and insertions/deletions (indels), which disrupt numerous coding variants. Although detecting putative deleterious mutations is straightforward, discerning the individual impact of these variants remains complex. However, advancements in understanding the context of these mutations may facilitate the training of machine-learning models to predict and rank their effects, ultimately benefiting plant and animal breeding and research on complex human diseases. The findings suggest that the occurrence of specific mutations induced by chemical mutagens can be anticipated, enhancing our comprehension of mutagen-induced genetic diversity and its phenotypic implications.

## Introduction

Mutagens can quickly generate novel, heritable genetic variation for identifying gene function when naturally occurring variation will not suffice. Sodium azide (NaN$_3$), ethyl methanesulfonate (EMS), and fast neutron (FN) radiation have been the most commonly used mutagens in plants. Sodium azide and EMS are chemical mutagens expected to cause point mutations [1]. FN is known for generating many large structural variants and small indels (especially deletions) that create frameshifts that disrupt gene function [2,3]. Mutagenizing agents have been used in many species to create knockouts or knockdowns of individual genes to understand gene function [4]. Despite decades of active use [5,6] and growing interest in the applications of mutagenesis [7,8], the characterization of the nature of the mutations generated has been limited [9]. More generally, the number of studies examining the effects of mutagens at the nucleotide sequence level is limited [1,3,10,11].

The nature and context in which mutations occur are essential determinants of their likely functional impact [12,13]. Recently reported efforts to employ sodium azide mutagenesis on a massive scale (500,000 mutated lines) clarify the need for an improved understanding of the number and types of mutations likely to be generated [7].

We can characterize induced mutations by comparing newly generated (*de novo*) mutations to variants occurring naturally in untreated lines. However, there are essential considerations to minimize false positives when distinguishing between induced mutations and existing variants. One challenge with identifying induced *de novo* changes is that the mutations are always a mixture of induced *de novo* and

spontaneously occurring mutations [13]. The experimental design necessary to study induced mutations in plant populations requires multiple generations of seed bulking before the mutagenesis treatment and several generations in which new mutations are made homozygous in inbred lines; thus, multiple generations in which new spontaneous variants can arise [3]. Mutagenesis experiments often use multiple generations of self-fertilization to ensure that mutations are meiotically heritable [14]. Mutagen dosage also affects the nature of observable mutations, with an effective dosage of mutagens defined by the LD50, or lethal dose for 50% of the treated sample, which results in the death of half the individuals. Of course, this creates significant attrition due to lethal mutations, resulting in some mutations or combinations that cannot be observed directly. Observable mutations are less damaging, allowing the plant to survive and reproduce.

New mutations, whether induced or naturally occurring, are more likely to be harmful than segregating variants subject to generations of natural (purifying) selection. More potentially harmful changes are observed among newer mutations in very deep resequencing panels [15] and induced mutations [3]. Newer mutations tend to include more changes of large effect, including nonsynonymous variants, changes in start and stop codons, intron splicing variants, and frameshifts [16,17]. Comparative approaches using phylogenetic constraint have been used for decades to predict which mutations are more likely to harm the organism [18,19]. More recent studies of induced mutations in *Arabidopsis thaliana* suggest that these approaches accurately identify phenotype-changing variants and are more likely to have a major effect on organismal fitness [20]. Thus, predicting the harmfulness of mutations has the potential to identify the nature of variants most likely to impact organismal fitness or yield in crops [21–23].

In barley, sodium azide primarily generates cytosine-to-thymine changes [24]. More recent studies have determined that mutagens create unique suites of mutations and that these mutations typically occur in a specific, immediate nucleotide context [1,13]. This observation is important because the nature of mutations, particularly when they occur predominantly in the presence of flanking motifs, can determine their relative effect. For example, C→T transitions, particularly in specific local sequence contexts, may limit the potential for mutagens to generate premature stop codons, even when the reverse complement of a mutation motif is considered. Olsen et al. 1993 [24] identified sodium azide mutations in the barley *Ant18* gene, finding that A.T→G.C base-pair transitions were most frequently generated. However, resequencing this single barley gene does not capture the effects of sodium azide at the whole-genome scale.

In the present study, we examine sodium azide-induced mutations in 11 mutagenized lines of Morex, the barley variety used as the primary reference genome [25]. We also generated whole-genome sequence for a sample of the Morex seed stock used for mutagenesis. In addition to single nucleotide changes, we find evidence of sequence insertions and deletions (indels) private to each mutagenized line. Mutations identified in sodium azide-treated lines were compared to variants in 13 barley landrace samples subject to whole-genome sequencing. To understand the nature of mutations induced by sodium azide, we address the following questions: 1) What is the nature of the variants induced by the mutagen? More specifically, does sodium azide tend to generate SNPs, indels, or other types of structural variants? 2) What is the observed mutation rate in sodium azide-treated samples in barley? 3) Do induced variants differ in type or predicted effect from variants occurring in untreated individuals? and 4) Is a greater number of harmful mutations associated with a reduction in yield in barley?

## Results & Discussion

### Identifying sodium azide-induced mutations

Data collected and analyzed in the present study includes three distinct datasets. First, we used multiple resequencing datasets to identify differences in our Morex samples and the Morex reference genome. This included new Illumina paired-end data with 10X Genomics linked reads, as well as both new and previously published Oxford Nanopore Technologies (ONT) and published Pacific Biosciences (PacBio) sequence [25] (Tables 1 and S1). A second data set included lines treated with sodium azide that were resequenced with Illumina paired-end data; for a subset of these lines we also generated linked reads and ONT DNA sequencing data (S1 Table). We used this data to identify single nucleotide and

**Table 1. Brief overview of sequencing datasets used in this study.**

| Acronym | Number Samples | Description | Reference | BioProject |
|---|---|---|---|---|
| 10X Genomics | 4 | Reads generated from 10X Genomics library preparation followed by 2x150bp PE Illumina whole genome sequencing for Morex and 3 mutated lines; 46X mapped coverage | This study | PRJNA849997 |
| WGS | 21 | 2x150bp PE Illumina whole genome sequencing | This study | PRJNA849997 |
| ONT | 4 | Oxford Nanopore reads of Morex and 3 mutated lines; 5X mapped coverage | This study | PRJNA967725 |
| ONT | 1 (100 seedlings) | Oxford Nanopore reads of Morex (sampled from 100 seedlings); 85X coverage | Mascher et al. 2021 [25] | PRJEB40588 |
| PacBio | 1 (100 seedlings) | PacBio circular consensus reads of Morex (sampled from 100 seedlings); 27X coverage | Mascher et al. 2021 [25] | PRJEB40587 |

structural variant differences between our Morex line published Morex_v3 reference genome [25]. We then identified mutations in lines subject to mutagenesis. For contrast between variants in mutagen-treated lines and naturally occurring variants, we used Illumina paired-end data to identify spontaneous variants in 13 barley landraces.

**Variants in our Morex sample relative to the Morex reference genome.** A major challenge in isolating *de novo* variants induced by the mutagen treatment is the need to distinguish between variants present in the mutagenized seed stock (i.e., Morex) and those that arise during mutagenesis. Heterogeneity, genetic variation within an inbred cultivar or variety, can contribute large numbers of variants [26] that are not due to the mutagen. Experimental contamination through unintended hybridization [3,27] can also contribute to large numbers of variants (S1 Fig). Mutagen-induced variants are expected to be relatively rare, requiring filtering of the variants that account for: (1) differences between the parental line used for mutagenesis and the reference genome, (2) uncallable regions (see Materials & Methods) that include genomic regions with unknown nucleotide state in the reference, or the regions where sequence reads do not align uniquely, and (3) heterogeneity among lines (S2 and S3 Figs). We identified callable portions of the genome that were 820 Mb and 817 Mb (the latter excludes low complexity sequence) relative to the 4.2 Gb genome size for Morex_v3 [25]. Callable regions capture 88% of high-confidence (HC) genes (31,625 HC genes out of 35,827 total HC genes) in the Morex_v3 reference genome. No individuals in our experiment show an excess of variants, runs of variants, or high heterozygosity consistent with recent hybridization [27].

Differences between the Morex_v3 reference genome and our Morex sample are mutations that have arisen in individual seed stocks, errors in the reference assembly, or errors in variant calling. To address these issues, we generated 10X Genomics-based resequencing with 46x and an ONT data set with 5x average mapped read coverage (S1 Table). We identified 52,596 SNPs, 8,203 1-bp indels (insertions and deletions), 3,182 indels ranging in size from 2-204 bp, and 53 deletions ranging from 41 bp - 60 Kbp that survived rigorous filtering (S2 and S3 Figs). Filtered ONT variant calls include 68 insertions and 42 deletions. Using published ONT (at 85x coverage) and PacBio data (at 27x coverage) sampled from 100 seedlings [25] (Table 1) that were incorporated in the Morex_v3 assembly, we detected an additional 178 insertions and 93 deletions that were excluded from callable regions.

**de novo variants in mutagenized barley lines.** Reference-based read mapping, variant calling in GATK [17,28,29], and filtering of the 11 $M_5$ mutagenized lines identified 23,339 SNVs (Single Nucleotide Variants) with an observed transition to transversion ratio (Ts/Tv) = 5.24. This compares to a ratio of Ts/Tv = 1.7 previously reported based on resequencing of naturally occurring variants in barley [17]. Among mutagenized lines, we also identified 5,376 smaller indels ranging in size from 1 - 296 bp for 28,715 variants potentially induced by the mutagen (Fig 1A). Here, we use SNV (Single Nucleotide Variants) to identify variants generated during mutagenesis and SNP (Single Nucleotide Polymorphisms) to identify variants segregating in the non-mutagenized population. Each variant is private to a mutagenized plant (i.e., occurs in only a single individual). There was an average of 2,122 SNVs and 489 indels per

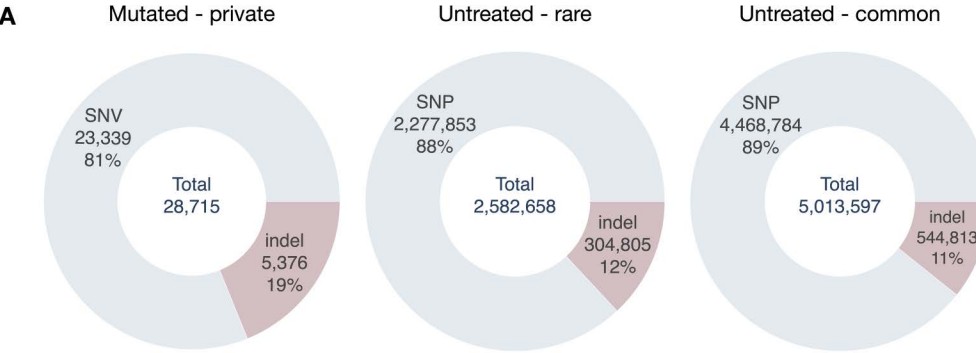

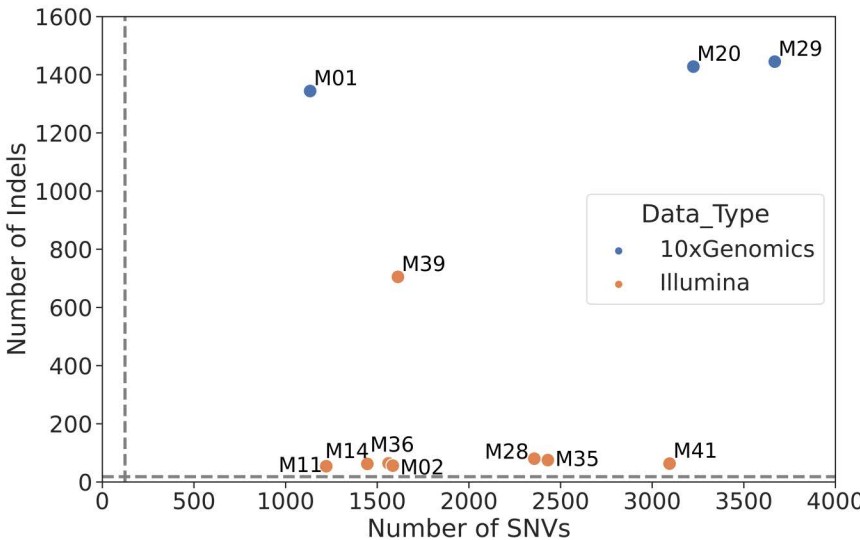

**Fig 1. A) The number and percentage of SNVs and indels in mutagenized, untreated rare, and untreated common categories. Private is defined as variants in mutated lines found in a single sample. Untreated are naturally occurring variants with rare variants defined as those with a non-reference allele count ≤ 2, while untreated common variants have alternate allele count ≥ 3. B) The number of SNVs and indels identified in each mutagenized line. Colors indicate whether 10X Genomics linked read technology was utilized for the sample. Dashed lines indicate the expected number of SNPs and indels based on average mutation rates and accounting for experimental design.**

mutated sample, with more indels identified in 10X Genomics samples utilizing linked reads (Fig 1B and Table 2). Those numbers are much higher (17x and 139x) than estimates of the average number of mutations that would spontaneously arise in the absence of a single-generation of sodium azide treatment. Those estimates were calculated as below (see Materials & Methods). Using the mean nucleotide substitution rate estimate of $6.5 \times 10^{-9}$ base substitutions per site per generation from [30] and accounting for our experimental design, we expect ~124 SNVs per individual in the 4.2 Gbp genome. For indels, we expect ~3.5 indels per individual to arise without the mutagen treatment based on an average indel mutation rate of $0.45 \times 10^{-9}$ for 1–3 bp indels and $0.5 \times 10^{-9}$ for >3 bp indels [30].

The ONT sequence and 10X Genomics linked reads of the same three mutagenized lines improved the detection of larger structural variants (SVs). No inversions passed the filtering criteria, and a set of high-quality duplications could not be identified; thus, structural variant calling focused on insertions and deletions. The SVs detected in the 10X Genomics data set included 52 deletions ranging from 41 bp to 60 Kbp (S4 Fig). ONT reads for three samples (S1 Table) were generated to validate the larger structural variants identified in the 10X Genomics data. The average ONT mapped read

**Table 2. Number of variants in each of the the sodium azide treated barley samples and the parent of the mutated lines (Morex-sample2). Counts for indels are variants private (appearing in only one sample) to each individual. Morex-sample2 is the same accession as the reference genome but derived from a different seed lot.**

| Accession | Type | Sequencing | SNV Count | Indel Count (1–296 bp) | Larger SVs (41 bp-60 Kbp) | | |
|---|---|---|---|---|---|---|---|
| | | | | | Total larger deletions (10X Genomics) | Total indels (ONT-Sniffles2) | Total indels (ONT-cuteSV) |
| Morex-sample2 | elite/ parent | 10X Genomics, ONT | 52,596 | 11,385 | 53 | 110 | N/A |
| M01 | mutated | 10X Genomics, ONT | 1,134 | 441 | 17 | 40 | 1 |
| M20 | mutated | 10X Genomics, ONT | 3,225 | 464 | 14 | 41 | 9 |
| M29 | mutated | 10X Genomics, ONT | 3,669 | 445 | 21 | 31 | 4 |
| M02 | mutated | Illumina 2x150bp | 1,585 | 56 | N/A | N/A | N/A |
| M11 | mutated | Illumina 2x150bp | 1,222 | 54 | N/A | N/A | N/A |
| M14 | mutated | Illumina 2x150bp | 1,446 | 62 | N/A | N/A | N/A |
| M28 | mutated | Illumina 2x150bp | 2,357 | 80 | N/A | N/A | N/A |
| M35 | mutated | Illumina 2x150bp | 2,431 | 75 | N/A | N/A | N/A |
| M36 | mutated | Illumina 2x150bp | 1,562 | 64 | N/A | N/A | N/A |
| M39 | mutated | Illumina 2x150bp | 1,613 | 705 | N/A | N/A | N/A |
| M41 | mutated | Illumina 2x150bp | 3,095 | 63 | N/A | N/A | N/A |

coverage was 3.4x (S1 Table). A total of 86 insertions (36–4,786 bp) and 26 deletions (36–300 bp) were called by Sniffles2; 8 insertions (21–172 bp) and 6 deletions (16–116 bp) were called by cuteSV (S4 and S5 Figs). These ONT calls provided direct sequence read-based confirmation of two larger deletion calls that were also called based on the 10X Genomics data set.

**Variants in untreated barley landraces.** For comparison, we generated whole-genome resequencing of 13 barley landraces [31]. Average coverage ranged from 41 - 93x (S1 Table), and after variant calling and quality filtering, we identified a total of 6.7 million SNPs with Ts/Tv = 1.74 and 849,618 indels ranging in size from 1 - 388 bp. Out of the 6,746,637 SNPs, 2,277,853 SNPs were categorized as rare (i.e., non-reference allele count of two or less, the allele was identified in one or two genotypes) with Ts/Tv = 1.71 and 4,468,784 were common (i.e., non-reference allele count of three or higher) with Ts/Tv = 1.76 (Fig 1A). Rare variants were compared to *de novo* variants in the treated lines because they have experienced fewer generations of selection, and their mutational spectrum is likely more similar to that in treated lines.

## Comparison of mutagenized versus untreated samples

SNPs in untreated samples are primarily transitions, particularly C→T* (Fig 2), where the notation C→T* includes the reverse complement G→A. Partitioning variants in the untreated samples into "rare" or "common" had a limited effect on the proportion of variants among each class, with a slight skew of rare variants to more C→T* transitions and fewer A→G*. Variants in sodium azide-treated lines were dominated by C→T* transitions, which comprised 79.1% of all SNVs in the mutagenized lines (Fig 2).

Single base pair changes dominate insertion and deletion variants, particularly in sodium-azide treated lines, constituting 28.3% of insertions and 36.4% of deletions (Fig 3). The pattern of indels in treated lines is similar to that in rare and common variants in standing variation. Notable differences include more 1- and 2-bp deletions and 1-bp insertions in treated lines (Fig 3). The 53 larger deletions (41 bp - 60 Kbp) called in the 10X Genomics data set represent roughly 1.7% of all 3,135 deletions in the mutated lines. Linked and long read sequencing was not possible for the landrace lines, precluding a direct comparison.

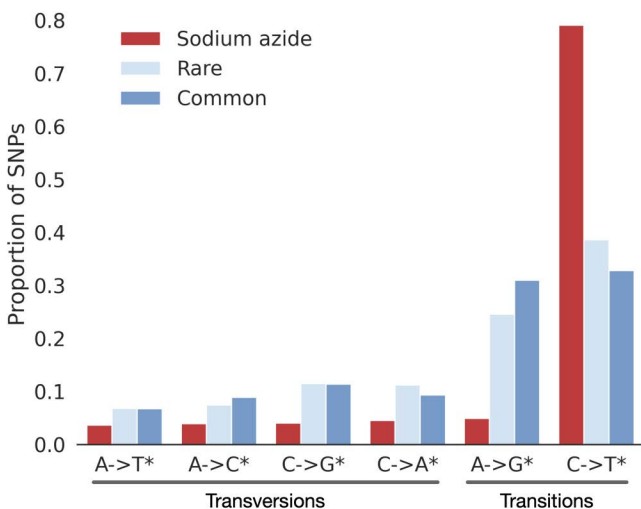

**Fig 2. The mutational spectrum of mutagenized, untreated rare, and untreated common SNPs.** Each bin includes the reverse complement. For example, the C→T* bin also includes G→A changes.

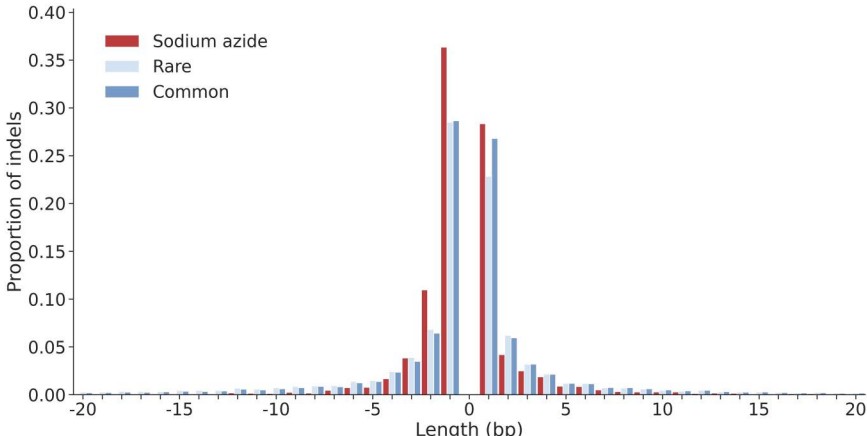

**Fig 3. The distribution of insertion and deletion lengths for sodium azide-treated lines versus rare and common variant categories.** Insertions are shown as positive values and deletions are shown as negative values. Only variants with lengths <20 bp are shown here.

In mutated samples, 9.7% of SNVs and 8.9% of indels occur in genic regions (S6 Fig). The percentage of variants in genic regions is lower in untreated lines for rare and common variants. Rare variants had 6.5% SNPs and 1.7% indels occurring in genic regions and are more directly comparable with mutated sample SNVs and indels. Variant Effect Predictor (VeP) [32] identified most sodium azide-induced variants as occurring in intergenic regions or genomic regions up or downstream of genes. A larger proportion of variants were found in intergenic regions among induced variants than in untreated lines. Fewer sodium azide-induced SNVs and indels were adjacent to genes (S7 and S8 Figs). Sodium azide-treated lines have a slightly higher proportion of missense variants (3.86%) than untreated lines (2.19% Rare, 2.32% Common), but this effect is small. Slight increases in the proportion of start-stop-related changes (0.2% Mutated, 0.09% Rare, and 0.07% Common) and splice donor and acceptor sites are also observed (0.09% Mutated, 0.01% Rare, and 0.01% Common). However, these variants are considered the most potentially damaging based on VeP categorization (S7

and S8 Figs). Larger deletion variants (41 bp - 60 Kbp) detected among the three lines with linked reads disrupt a genic region in 11.3% of cases (6 out of 53 total).

## Harmful mutations based on phylogenetic constraint

On average, sodium azide-treated lines include 78.6 nonsynonymous SNVs per sample, with 865 nonsynonymous SNVs identified among the 11 mutated lines (S9 Fig). Estimates of putative variant effects based on phylogenetic constraints [20] were used to identify potentially damaging nonsynonymous variants among primary transcripts in the barley genome. This analysis includes missense variants (a change in amino acid), start lost, stop gained, and stop lost variants based on the Sequence Ontology definition of nonsynonymous changes [33]. For the 11 mutated lines, 611 nonsynonymous mutations in primary transcripts were tested for phylogenetic constraint relative to 72 other angiosperms to identify potentially universally harmful mutations. Among the successfully annotated mutations, 155 (35.9%) were annotated as "harmful" (i.e., putatively deleterious), while the remaining 277 (64.1%) were identified as "tolerated." This value compares to 9,716 "rare" nonsynonymous variants tested in 13 barley landraces, where 1,633 (13.2%) are identified as "harmful" and 10,693 (86.8%) as tolerated. For "common nonsynonymous variants, 14,537 were tested, where 23,506 (7%) are "harmful," and 311,121 (93%) are tolerated.

An average of 14.1 (± 6.46) deleterious SNVs were identified per mutagenized sample (S10 Fig). Given that these changes at conserved coding positions are frequently phenotype-changing [20], this suggests roughly 14 disrupted SNVs per individual treated line. The ratio of nonsynonymous SNVs (nSNV) to synonymous SNVs (sSNV) in mutated lines is 1.8:1. In comparison, the ratio of nSNVs:sSNVs in rare and common SNP categories is 1.36:1 and 1.22:1, respectively. The proportion of nSNVs inferred to be deleterious was 17.9% in treated lines versus nSNPs at 3.2% in rare and 4.1% in common categories. To standardize results among samples, we identified the number of harmful mutations per codon in 10 Mbp windows. The proportion of dSNPs per codon was lower near the centromeres for rare and common SNPs in the landraces (S11-S13 Figs).

## The context of variants induced by sodium azide

Biochemical interactions between mutagenic compounds and DNA produce SNVs in specific nucleotide contexts [13]. We used the program Mutation Motif [9] for all SNVs to examine this effect in sodium azide-treated barley lines. The predominant mutation types in both treated and untreated lines are C→T* changes. In sodium azide-treated lines, the cytosine that changes to thymine is frequently followed by another cytosine, creating a CC context of mutation on the forward strand (Fig 4). To our knowledge, there have not been previous studies on the preferential context of sodium azide mutations. There are highly significant differences between sodium azide-induced variants and variants spontaneously originating in the genome (S2 Table). In untreated lines, the mutated cytosine is generally followed by guanine at the +1 or +2 site (downstream) from the C, thus resulting in a CG or potentially CGG context in which mutations occur. In the complete 4.2 Gb Morex_v3 genome assembly, the CC, CG (the two bp motif for CpG changes), and CGG motifs occur ~228 million, ~154 million, and ~40 million times, respectively. In the 820 Mb region in which unique single nucleotide variants could be called, CC occurs ~48 million times, CG occurs ~35 million times, and CGG motifs occur ~9 million times. This suggests that, on average, a single generation of sodium azide treatment resulted in the mutation of 0.013% of CC sites at which unique mutations could be detected. The CC and CG motifs constitute 5.8% and 4.3% of all two nucleotide combinations in the 820 Mb callable regions, and the CGG motif constitutes 1.1% of three nucleotide combinations. In contrast, AA and TT motifs are the most frequent two nucleotide motifs, making up 8.1%.

Amino acid changes identified in sodium azide-treated lines are dominated by those that include the CC or (reverse complement) GG motif (S14 Fig). Glycine to aspartic acid, proline to serine, and alanine to threonine are the three most abundant amino acid changes in SNVs identified as harmful (i.e., deleterious) (Table 3 and S15 Fig). The top four changes in tolerated SNVs (tSNVs) in mutagenized samples are similar to those annotated as harmful; tolerated amino acid

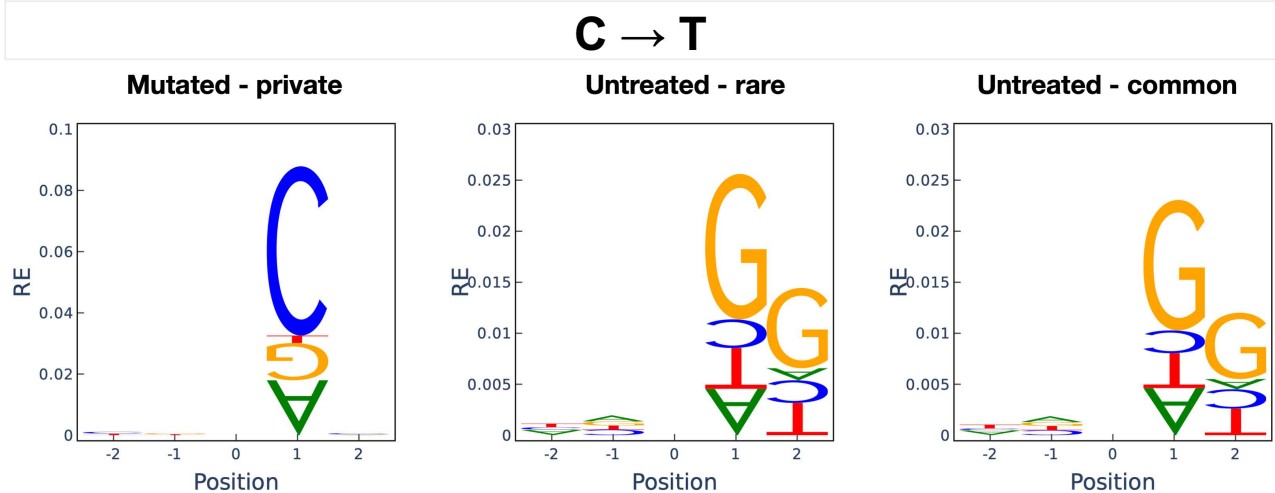

**Fig 4. The nucleotide sequence context for C →T transitions relative to the reference genome in the mutated (*N* = 10,048), untreated rare (*N* = 439,563), and untreated common (*N* = 732,565) SNVs.** Position 0 indicates where the C→T change occurred. The relative height of the letters indicates their relative entropy (RE), with a higher RE indicating a position has a greater influence on the mutation. Upright letters indicate overrepresented bases, whereas upside-down letters indicate underrepresented bases at positions neighboring position 0. The null expectation (RE of zero) is based on randomly sampling a nearby location with the same starting base (e.g., for a C→T mutation, a random choice of a position with a C is selected.

changes include alanine to threonine, alanine to valine, glycine to aspartic acid, and proline to serine (Table 3). This contrasts with amino acid changes induced by rare and common variants in standing variation, where transitions associated with CpG are more abundant. For rare and common dSNPs, alanine-to-threonine and alanine-to-valine changes appear at the highest frequencies. The arginine-to-cysteine amino acid change had the third highest frequency in the common dSNPs class and frequently annotates as deleterious.

## Putatively harmful SNVs and phenotypic variation

A total of 25 mutagenized barley lines self-fertilized for five to seven generations ($M_{5:7}$) were used for yield testing at one location in the first year (St. Paul, MN) and three locations in Minnesota (Crookston, Lamberton, and St. Paul) in years 2 and 3. Yield testing was performed in the presence of 5–8 check lines (see Materials and Methods) and the original Morex line untreated with the mutagen. Data for heading days after planting (DAP) and plant height were also collected. After spatial adjustment for variation across plots, the average yield for each line was calculated for all years combined. As expected, most mutagenized lines had lower grain yields than the Morex W2017 parental line, with six mutagenized lines yielding roughly the same or slightly higher than the parental line (Fig 5). M29 is among the three lowest-yielding lines and is the only mutagenized line with a visibly distinct phenotype, described as onion-like, short-stature, and very compact (S16 Fig). Mutagenized lines tend to have diminished yield relative to Morex, though some line-by-year combinations slightly exceeded the yield in Morex and some checks (S17 Fig). The average diminution in yield across years and lines for the mutagenized lines was 32.8% relative to the Morex W2017 parent. The heading DAP in mutagenized lines increased by 6.2%, and height was reduced by 15% compared to the Morex W2017 parent line. To determine if observed damaging mutations impacted yield, we compared the relative order of yield to the number of damaging mutations per line. We found a slightly negative but nonsignificant correlation of -0.28 (*P* = 0.4) between the number of harmful variants and yield (S3 Table). Most mutagenized lines had lower variance across replicates than the check lines. This is likely due to the experimental design with seeds originating from plants that can be traced through single-seed descent following the mutagen treatment, whereas check lines derive from more heterogeneous seed stocks.

**Table 3. Most abundant amino acid changes annotated as deleterious (DEL) and tolerated (TOL) for mutated, rare, and common SNPs.**

| | Mutant | | | Rare | | | Common | | |
|---|---|---|---|---|---|---|---|---|---|
| | AA Change | Number | % | AA Change | Number | % | AA Change | Number | % |
| DEL SNVs | G/D | 22 | 14.19 | A/T | 62 | 3.97 | A/T | 140 | 3.18 |
| | P/S | 20 | 12.90 | A/V | 61 | 3.90 | A/V | 137 | 3.12 |
| | A/T | 19 | 12.25 | G/D | 38 | 2.43 | R/C | 91 | 2.07 |
| | A/V | 10 | 6.45 | P/L | 37 | 2.37 | L/F | 86 | 1.96 |
| TOL SNVs | A/T | 40 | 14.44 | A/T | 395 | 3.89 | A/T | 1632 | 3.29 |
| | A/V | 33 | 11.91 | A/V | 383 | 3.77 | T/A | 1590 | 3.21 |
| | G/D | 23 | 8.30 | V/I | 265 | 2.61 | A/V | 1552 | 3.13 |
| | P/S | 23 | 8.30 | T/A | 264 | 2.60 | V/A | 1548 | 3.12 |

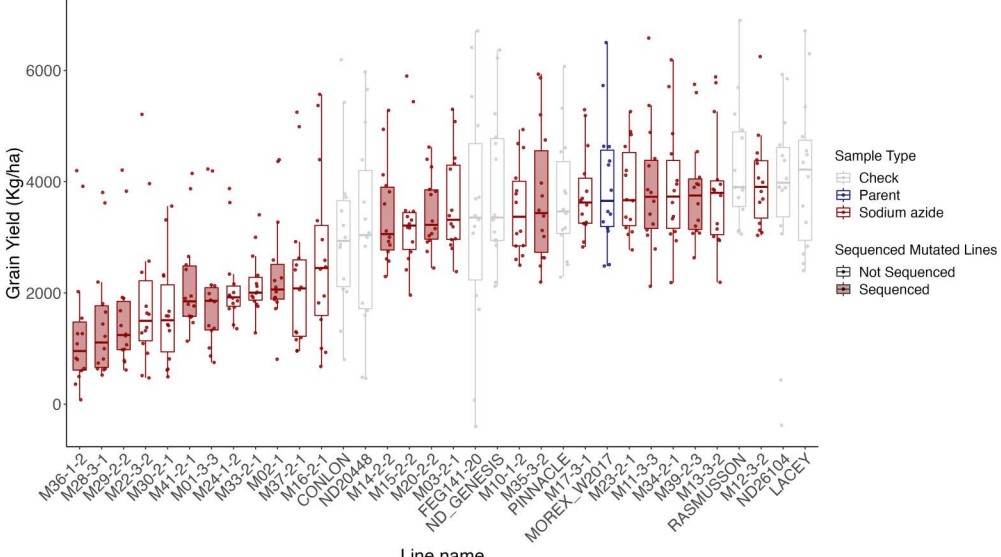

**Fig 5. Grain yield for 25 mutated lines, the Morex W2017 parent, and 8 check lines.** The box plots are sorted by the median for each line, and the bars in the box plot indicate the mean. Sodium azide-treated lines are represented by red outlines. Red shaded boxes indicate mutated lines that were sequenced in this study.

## Conclusions

Sodium azide is widely used as a mutagen in experimental plant populations. It has been frequently used for inducing variants in barley [24,34], including recent reports of extremely large-scale experiments involving the characterization of hundreds of thousands of individual plants [7]. Other mutagens used historically have included X-rays, neutrons, ethylene imine, sulfonates, other chemical mutagens, and various combinations of mutagens [detailed in 35]. However, most studies have focused on the phenotypic effects of mutagenesis [4,36,37] or changes induced at individual genes [24]. The genome-level effects of the mutagen have rarely been examined.

Consistent with a prior single gene resequencing study, we find that C→T* transitions dominate induced mutations [7,24,38] (Fig 2). A similar pattern of nucleotide change has been found for ethyl methanesulfonate (EMS) induced mutation [10,39]. N-ethyl-N-nitrosourea (ENU) induced mutations in mice produce primarily A→G* or A→T* mutations [13]. In

contrast, C→T* transitions predominate among the variants observed after fast neutron treatment, but with less enrichment relative to other SNVs [3,10].

Sodium azide appears to generate single nucleotide variants primarily. We identify an average of 2,122 SNVs per mutagenized line. This is an ~88-fold increase in SNVs compared to expectations in the absence of a single-generation sodium azide treatment (see Materials and Methods for equations 1–6). Induced indels of all sizes are less abundant (Figs 1 and 3) but occur at ~130–140 fold higher rate than nominal mutation rates.

The observation of higher indel rates derives from the comparison of data from multiple sequencing platforms, including linked-reads and long-read sequencing, with an average of 489 indels per line in mutagenized $M_5$ lines. The mutations present in $M_5$ lines are necessarily a mixture of induced mutations and mutations that arose spontaneously during line maintenance [13]. However, based on mutation accumulation resequencing studies in *Arabidopsis thaliana*, the 1–3 bp mutation rate was estimated as an average of 0.45 x 10-9 indels per site per generation, and the large deletions (>3 bp) mutation rate was estimated as 0.5 x 10-9 [30]. In the 820 Mb portion of the barley genome, where variants can be called unambiguously, we expect ~3.5 indels per individual to arise naturally without mutagenesis treatment over the course of the experiment (see Methods for equations 1–6). In the 4.2 Gbp barley genome, we would expect ~18 indels per individual (Fig 1B). This is a ~130–140-fold increase in the indel mutation rate of treated lines to an average of 5.13 x 10-7 indels per site per generation.

Most sodium azide-induced mutations occur in a specific nucleotide sequence context, as C→T* changes in a CC mutation motif (Fig 4) or the reverse complement. This results in a relatively small number of amino acid changes that predominate among induced mutations. Li et al. 2017 [10] describe similar results for mutations found after EMS or EMS/ENU treatment, where the most common changes within codons produce only a fraction of all possible amino acid changes. In a similar manner, sodium azide-induced amino acid changes are very distinct from most amino acid changes segregating in barley. The amino acid changes that annotate as harmful and predominate in the mutagenized samples are glycine to aspartic acid, proline to serine, and alanine to threonine. In comparison, the top three amino acid changes annotated as harmful and segregating in the untreated barley landraces include alanine to threonine, alanine to valine, and arginine to cysteine. For projects seeking to induce novel changes, for example, in disease resistance genes or genes associated with stress tolerance, sodium azide will induce many coding changes that are rarely observed among standing variation.

Resequencing of individual genes identified many sodium azide-induced SNVs in barley [24]. Induced indels and SVs were not previously reported but would be difficult to identify with Sanger sequencing. Indeed, sequencing technology continues to present a limitation. Many of the SVs identified here were identified by linked reads (in two cases verified by ONT long reads) but could not be identified by Illumina paired-end reads alone. Regarding relative effect, indels and SVs identified with linked reads and verified with ONT result in six disruptive mutations that either induce a frameshift or eliminate a portion of a coding gene. This results in an average of two structural disruptions of genes per individual instead of an average of 33.4 per individual due to 1–3 bp nucleotide sequence-level changes.

Our mutated lines average a 37.7% reduction in yield relative to their non-mutated parental Morex line. This reduction in yield can be attributed to an average of 14.1 induced SNVs and 37.3 indels per line. The typical line has an average of 249.1 disruptions of coding variants, including SNVs and indels (S6 Fig). The approach used in this study identified a finite number of deleterious (i.e., harmful) mutations induced by sodium azide. It was successful at creating lines that had lower yield than the untreated Morex parent line in the experiment. The reduction in fitness (using yield as a proxy for fitness) following the mutagen treatment was expected, given that most new amino-acid-changing mutations that impact fitness will be deleterious [16,40].

In practical applications, deleterious variants are relatively easy to detect, which makes it possible to select against them or eliminate them via targeted replacement of individual variants [22,23,41,42]. However, it is still challenging to identify the effects of individual deleterious variants. With lines that have reduced yield and a better understanding of the nature of changes generated by sodium azide and the sequence contexts in which they occur, there is the possibility of

training machine-learning models to predict which variants contribute to harmful phenotypic change [43,44]. Then, it will be possible to rank the expected effect size of each harmful variant and combine the predictions with existing genomic prediction approaches to benefit plant and animal breeding programs [45] and the study of complex human diseases. All of these applications require a means to identify the most harmful variants.

## Materials and methods

### Plant materials and mutagenesis

Barley from a Morex seed stock was treated with sodium azide following the protocol in [46]. Morex is a 6-row malting barley variety used as the primary reference genome [25]. Morex was chosen for these experiments to facilitate the identification and isolation of *de novo* variants because the reference makes it easier to distinguish between variants that are different between our parent sample and the reference genome versus variants that arose from the mutagen. The Morex line in this experiment traces back to the parent seed stock used to generate the Steptoe x Morex doubled haploid barley mapping population [47]. To generate sufficient Morex seeds for sodium azide mutagenesis, 120 seeds were planted from a single Morex plant (S1 Fig). Next, 200 seeds from the resulting bulk of seeds were planted; this was repeated one more time. A portion of the seeds was treated with sodium azide (1 mM $NaN_3$) following the [46] protocol; the remaining portion of untreated Morex seeds was planted for another round of seed bulking and then planted to collect leaf tissue for sequencing (S1 Fig). After mutagenesis, the resulting mutagenized seeds were grown to maturity and harvested, forming the $M_1$ generation. These individuals then underwent single-seed descent until $M_5$.

Estimates for the expected number of spontaneous mutations occurring without mutagen treatment were calculated using the experimental design for seven generations of self-fertilization (S1 Fig) and rate estimates from [30]. The mutation rates used for SNPs was $6.53 \times 10^{-9}$ and for 1–3 bp indels was $0.45 \times 10^{-9}$. For the 820 Mbp (820,594,305 bp) callable regions, a diploid genome size of 1,641,188,610 bp was used. For estimates of the 4.2 Gbp (4,225,605,719 bp) genome size [25], a diploid genome size of 8,451,211,438 bp was used. Each generation, new mutations appear in the heterozygous state, and the number of new heterozygous mutations ($N_{het}$) is given by

$$N_{het} = Diploid\ bp * Mutation\ rate \tag{1}$$

The experimental design involved multiple generations of selfing, meaning *de novo* mutations from previous generations are being lost or fixed over time. In each generation, heterozygous mutations are inherited ($I_{het}$) and are represented by

$$I_{het} = \begin{cases} 0 & for\ x = 0 \\ 0.5 \times N_{het}G_{x-1} & for\ x = 1 \\ 0.5 \times N_{het}G_{x-1} + 0.5 \times I_{het}G_{x-1} & for\ x > 2 \end{cases} \tag{2}$$

where *x* is the current generation. Similarly, each generation homozygous mutations are inherited ($I_{hom}$) and are represented by

$$I_{hom} = \begin{cases} 0 & for\ x < 2 \\ 0.25 \times I_{het}G_{x-1} & for\ x = 2 \\ I_{hom}G_{x-1} + 0.25 \times I_{het}G_{x-1} & for\ x > 2 \end{cases} \tag{3}$$

The number of heterozygous and homozygous spontaneous mutations accumulated by generation seven in our experimental design is given by

$$Total_{het}G_7 = I_{het}G_7 \tag{4}$$

$$Total_{hom}G_7 = I_{hom}G_7 \tag{5}$$

$$Total_{spont}G_7 = I_{het}G_7 + I_{hom}G_7 \tag{6}$$

and is used as our estimated number of spontaneous mutations that would have been present without mutagen treatment.

## Phenotypic data collection

Twenty-five $M_{5:7}$ mutated lines and the $F_{1:2}$ W2017 Morex parent line were evaluated in yield trials with two replicates at one location (St. Paul, MN) in 2020 and three locations in Minnesota (Crookston, Lamberton, and St. Paul) in 2021 and 2022. Lines were grown in a randomized complete block design. Phenotypic data on grain yield, heading days after planting (DAP), height, and lodging were collected. Check varieties were used to adjust for spatial variation across trial plots for traits with a continuous scale (yield, heading DAP, and height). Spatial adjustments were performed using the R package mvngGrAd [48]. Check lines were chosen because of well-quantified field performance and included Conlon, FEG141–20, Lacey, ND20448, ND26104, ND_Genesis, Pinnacle, and Rasmusson.

## Whole-genome short-read and long-read sequencing

We generated whole-genome sequencing in 25 barley (*Hordeum vulgare* ssp. *vulgare*) accessions: the parent of the mutagenized lines (W2017 Morex), 11 mutagenized lines, and 13 barley landraces [31] for comparative analyses (Tables 1 and S1). High molecular weight genomic DNA was extracted from 4-6 week-old leaf tissue collected on ice using the Cytiva Nucleon PhytoPure kit for the mutagenized lines. We sequenced three of the 11 $M_5$ mutagenized lines (M01, M20, and M29) and the W2017 Morex line using 10X Genomics linked read library preparation followed by Illumina NovaSeq 6000 sequencing with 150-bp paired-end technology to a target depth of 40x. For the remaining eight $M_5$ mutagenized lines (M02, M11, M14, M28, M35, M36, M39, and M41), libraries were prepared using Illumina DNA Prep followed by Illumina NovaSeq 6000 sequencing with 150-bp paired-end technology to a target depth of 16x. Sequences for the 13 landraces were generated using Illumina TruSeq DNA Nano Prep followed by Illumina NovaSeq 6000 sequencing with 150-bp paired-end technology to a target depth of 40x.

We used Oxford Nanopore Technologies (ONT) to sequence W2017 Morex and three of the 11 $M_5$ mutagenized lines (M01, M20, and M29, see Tables 1 and S1). This data was collected to provide read-level confirmation of SVs indicated by the Illumina short-read resequencing. For sample M01, following the ONT protocol, a high molecular weight gDNA extraction was performed using the Qiagen Genomic-tip kit (10262) with Carlson Lysis buffer (10450002–1). High molecular weight gDNA extractions for samples M20 and M29 were generated using the NucleoBond HMW DNA kit (740160.20) from Takara Bio USA. Size selection was performed using Circulomics SRE buffer, and DNA was quantified using the Qubit assay. The libraries were prepared with 400 ng of gDNA using the Rapid Sequencing Kit (SQK-RAD004) following the protocol version RSE_9046_v1_revT_14Aug2019. The library was primed using the flow cell priming kit (EXP-FLP002), then 400 ng of the library was loaded onto an R9.4.1 flow cell (FLO-MIN106D). M01 was run for 72 hours on a MinION Mk1C (MIN-101C). We found the flow cells were no longer collecting new data after 24 hours and modified the run for the remaining two samples. For M20 and M29, 400 ng of the library was loaded onto an R9.4.1 flow cell, run for 12 hours, and then paused. At this point, the pores were cleared using the flow cell wash kit (EXP-WSH004), then 400 ng of additional library was loaded and run for another 12 hours before the run was paused. Again, pores were cleared with the flow cell wash kit, then 400 ng of an additional library was loaded. The flow cell was run for an additional 24–30 hours. Three reactions were run for a single flow cell; a single sample with two washes ran for a total of 48–56 hours. This approach produced the highest data output for our samples. This process was repeated until each mutagenized line was

sequenced to a target depth of 2-3x (S1 Table). Basecalling was performed using Guppy v5.0.12 + eb1a981 (for all runs except one) and Guppy v5.0.17 + 99baa5b (for one M01 run #3) using the default setting on the MinION Mk1C.

**Read mapping and variant calling**

Read alignment and variant calling for the eight mutagenized and 13 landrace WGS lines were processed using the sequence_handling workflow (https://github.com/MorrellLAB/sequence_handling), which integrates publicly available software into a series of bash scripts [49]. The configuration files, which identify software versions and parameters, and scripts are available in the GitHub repositories https://github.com/MorrellLAB/hybrid_barley and https://github.com/Morrell-LAB/Barley_Mutated. Reads were aligned against the third version of the barley Morex reference genome (Morex_v3) [25] with parameters adjusted to account for the level of nucleotide diversity in barley. Variants were called as part of a larger set of samples and followed the Genome Analysis Toolkit (GATK) best practices recommendations [28,29]. SNPs underwent GATK VariantRecalibrator with the following as input: filtered variants and SNPs from genotyping assays, which include 2,975 BOPA SNPs [50], 7,541 9K SNPs [51], and 41,813 50K SNPs [52]. Only polymorphic and biallelic SNPs were included. Additional SNP filtering criteria include allele balance [53,54] deviation of 0.1, proportion heterozygous genotypes at a site > 0.1, per sample minimum DP < 5, per sample maximum DP > 158, proportion missing genotypes at a site > 0.30, QUAL < 30, and GQ < 9. SNPs identified in a barley Sanger resequencing dataset [55,56] were used for validation. Indels were filtered following GATK's Best Practices Guidelines for hard filtering since we did not have enough truth and training datasets to run indels through VariantRecalibrator. All filtering criteria are detailed in the scripts available in the GitHub repository https://github.com/MorrellLAB/Barley_Mutated.

For the four 10X Genomics samples (W2017 Morex and three mutagenized lines, see S1 Table), reads were aligned to the Morex v3 reference genome, and variants were called with the 10X Genomics software, Long Ranger v2.2.2. The Long Ranger pipeline processes the Chromium-prepared sequencing samples. Variants were filtered based on filters generated by Long Ranger, which include: 10X_QUAL_FILTER, 10X_ALLELE_FRACTION_FILTER, 10X_PHASING_INCON-SISTENT, 10X_HOMOPOLYMER_UNPHASED_INSERTION, 10X_RESCUED_MOLECULE_HIGH_DIVERSITY, and LOWQ. SNPs and 1-bp indels were filtered to sites with per sample DP between 5 and 78 and an allele balance deviation of +/- 0.2 (from the expected 0.5) for heterozygous genotypes.

For the ONT data of W2017 Morex, M01, M20, and M29, read quality and summary statistics were generated with NanoPlot v1.38.1 and pycoQC [57]. Adapters were trimmed with Porechop v0.2.4 [58]. Reads were then aligned to the barley Morex v3 reference genome using Minimap2 v2.17 [59] with parameters recommended for ONT sequence reads. The resulting SAM files were then realigned using a modified version of the Vulcan pipeline [60] (customized version, https://gitlab.com/ChaochihL/vulcan), which utilizes NGMLR [61] for read realignment, converted to BAM format, and sorted using Samtools v1.9 [62]. Structural variants were then called using Sniffles v2.0.3 [61,63].

We also used a publicly available Morex data sequenced with PacBio CCS reads (BioProject PRJEB40587, ERR numbers ERR4659245-ERR4659249) [25]. We used HiFiAdapterFilt [64] to filter adapters. Reads were aligned using Minimap2 v2.17 [59] using parameters recommended for PacBio sequence reads. Similar to the ONT data, the resulting SAM files were realigned using a modified version of the Vulcan pipeline [60] (customized version, https://gitlab.com/ChaochihL/vulcan), converted to BAM format, and sorted using Samtools v1.9 [62]. Structural variants were called used Sniffles v2.0.3 [61,63].

For all datasets in this study, mapped coverage was estimated using Mosdepth v0.3.1 [65]. All mapping parameters and filtering criteria are detailed in scripts available in the GitHub repository (https://github.com/MorrellLAB/Barley_Mutated).

## Identifying *de novo* variants

**Part 1: Finding differences between Morex samples and Morex reference genome.** To identify *de novo* variants induced by the mutagen, we generated a list of regions where variants were called in Morex-sample2 (W2017 Morex); these are differences between the Morex parent in this study and the Morex reference genome. Variants called in the [25] Morex ONT and PacBio data were also counted as differences from the reference; these are potentially due to heterogeneity (variation among individuals in the Morex variety). To minimize spurious SV calls in difficult-to-call regions for the 10X Genomics, ONT, and PacBio data, we filtered out variants that overlap with uncallable regions, which includes annotated repeats, stretches of N's in the reference genome sequence, and "high copy" regions (i.e., regions where plastids, rDNA repeats, and centromere repeats align). For the ONT and 10X Genomics data, SVs that overlap low complexity regions (defined as regions containing low-copy sequence) were also filtered out because they can be non-biological artifactual sequences that result in unmapped sequences or sequences mapped to multiple locations. Low complexity regions were generated using BBMask from BBTools (BBMap – Bushnell B. – sourceforge.net/projects/bbmap/) with entropy set at 0.7, which was determined through data exploration to capture a majority of low complexity sequences (scripts available at https://github.com/MorrellLAB/morex_reference/tree/master/morex_v3). For the ONT and PacBio data, SVs were filtered out if they had less than five supporting reads. For Morex-sample2, SNPs in 100 bp windows with >2% diversity were filtered out. Such high diversity windows are unlikely when aligning Morex-sample2 to the Morex reference genome. SVs were then visually inspected in IGV or scored in SV-plaudit (in the case of the 10X Genomics larger deletions), and filtering criteria were tuned if necessary (summarized in S2 Fig). The filtered SVs form a high confidence set of places where *de novo* variants shouldn't be called due to regions that are difficult to align or are heterogeneous among Morex individuals.

**Part 2: de novo filtering mutated individuals.** For the 10X Genomics and ONT sequenced mutagenized lines (M01, M20, M29), SVs that overlap uncallable regions or low complexity regions were filtered out (S3 Fig). To benefit from using the strengths of distinct SV callers for ONT data, we utilize Sniffles2 [60] and cuteSV [66]. SVs called by cuteSV and Sniffles2 in the ONT data were filtered similarly, except in the Sniffles2 calls, we required at least five supporting reads. For all sequenced mutagenized lines (10X Genomics, ONT, and Illumina WGS), variants that overlap the "differences from reference" regions were filtered out (summarized in S2 Fig). SNPs identified in the mutagenized samples that also appear in the BOPA (Barley Oligo Pooled Assay 1 and 2) on the Illumina Golden Gate genotyping platform [50], Barley 9K Illumina Infinium iSelect Custom Genotyping BeadChip [67], and Barley 50K iSelect SNP array [52] panels were also excluded. Variants were filtered to those private to individual mutagenized samples, meaning the variant only exists in one of the mutagenized samples at a genomic position. This is based on the expectation that variants induced by the mutagen are new (arose after the mutagen treatment was applied) and unique to each individual. So, variants identified in the mutagenized lines that also exist in the Morex samples are likely due to heterogeneity in Morex and were not generated by the mutagen treatment. Again, variants were visually inspected in IGV [68] or igv-reports (https://github.com/igvteam/igv-reports), and filtering criteria were tuned if necessary.

An image scoring approach was used to verify the larger deletions in the 10X Genomics three mutagenized samples. Images of the SVs were created with the SamPlot software [69]. Following the pipeline implemented in SV-Plaudit [70], the SV images were stored in the Amazon Web Services cloud storage and scored by multiple investigators through the PlotCritic website (https://github.com/jbelyeu/PlotCritic) based on the following criteria: coverage, insert size, and linked/split read evidence. This produced a set of scored deletions where a majority of scorers confirmed read evidence for the variant. This confirmed variant set was then verified using the ONT-sequenced samples (M01, M20, and M29).

## Nucleotide composition of variants

Mutation Motif [9] was used to identify the most frequent sequence motifs that affect SNPs in the mutated, rare, and common variant classes. This program performs comparative statistical analyses of neighborhoods (5 bp windows) centered

around each SNP to identify the frequency of sequence motifs and the influence of neighboring bases for each SNP. The neighborhood of each SNP (e.g., C→T mutation) is compared to the neighborhood of a reference occurrence of the same nucleotide (e.g., C) randomly sampled from within +/- 100 bp of the mutation. Comparing the motifs associated with mutated, rare, and common classes of SNPs allowed us to identify specific motifs that sodium azide may preferentially target. The number of 2- and 3- bp motifs in the 4.2 Gb reference genome and 820 Mb callable regions was calculated using the EMBOSS [71] compseq tool.

### Deleterious predictions

Ensembl Variant Effect Predictor (VeP) [32] was used to determine the predicted effect of each variant in the filtered VCF file, which includes SNPs, insertions, and deletions. Identification of nonsynonymous variants (includes missense, start lost, stop gained, and stop lost variants) used gene models provided by [25]. Nonsynonymous variants for the mutated samples and landraces were extracted from the VeP reports and assessed using BAD_Mutations [17,20], which includes a likelihood ratio test [72] that compares codon conservation across Angiosperm species to determine if a base substitution is likely to be deleterious. We ran the BAD_Mutations pipeline with a set of 72 Angiosperm species genome sequences that are available through Phytozome v13 (https://phytozome-next.jgi.doe.gov/, last accessed November 29, 2021) and Ensembl Plants (http://plants.ensembl.org/, last accessed November 29, 2021). We ran BAD_Mutations using 35,827 primary transcripts. A SNP was annotated as deleterious if the *P*-value for the test was < 0.05 with a multiple tests correction based on the number of tested codons, minimum of 10 sequences, maximum constraint of 1, and if the alternate or reference allele was not seen in any of the other species. Our thresholds for the three data groups were 8.1E-5 (611 codons tested) for the mutated samples, 5.1E-6 (9,716 codons tested) for the rare variants, and 3.4E-6 (14,537 codons tested) for the common variants. SNPs that failed this set of criteria were annotated as tolerated.

### Supporting information

**S1 Fig. Experimental design for generating the sodium azide treated lines.** $G_x$ is the generation used for spontaneous mutation estimates (see Methods).
(TIFF)

**S2 Fig. Flowchart of part 1 of variant filtering to identify differences between the Morex parent of the mutagenized lines and the Morex reference genome.** These are regions where induced mutations should not be called as they are more likely to be variation among Morex individuals.
(TIFF)

**S3 Fig. Flowchart of part 2 of variant filtering to identify *de novo* variants induced by sodium azide.** Larger SVs were visually evaluated with a similar approach as in part 1 of the filtering (S2 Fig).
(TIFF)

**S4 Fig. The distribution of larger deletion sizes in three mutagenized lines (M01, M20, and M29) as called by 10X Genomics Longranger, Sniffles2 (ONT), and cuteSV (ONT).**
(TIFF)

**S5 Fig. The distribution of larger insertion sizes in three mutagenized lines (M01, M20, and M29) as called by Sniffles2 (ONT) and cuteSV (ONT).** Larger insertions were not called in the 10X Genomics dataset.
(TIFF)

**S6 Fig.** A) A summary of the percentage and number of SNPs and indels in genic regions that can potentially disrupt genes for sodium azide-treated samples and rare vs. common categories. B) Per sample breakdowns of the number and percentage of SNVs and indels that are in genic regions and can potentially disrupt genes.
(TIFF)

**S7 Fig. The functional effects of mutagenized, untreated rare, and untreated common SNVs/SNPs as annotated by VeP. Bars are labeled with each consequence type's percentage and number of SNVs/SNPs.** Boxes on the left side indicate the impact classification of the consequence type.
(TIFF)

**S8 Fig. The functional effects of mutagenized, untreated rare, and untreated common indels as annotated by VeP. Bars are labeled with the percentage and number of indels in each consequence type.** Boxes on the left side indicate the impact classification of the consequence type.
(TIFF)

**S9 Fig. The number of nonsynonymous and synonymous SNVs in each mutated sample.**
(TIFF)

**S10 Fig. The number of nonsynonymous SNVs in each mutated sample was partitioned into "Deleterious" and "Tolerated. ".**
(TIFF)

**S11 Fig. The number of nonsynonymous SNPs per covered codon in 10 Mb windows in mutated samples across the barley genome.** Nonsynonymous SNPs are separated into "Deleterious" vs. "Tolerated" and are plotted separately. The vertical grey line indicates the centromeric region.
(TIFF)

**S12 Fig. The number of nonsynonymous SNPs per covered codon in 10 Mb windows categorized as "rare" in the landrace samples across the barley genome.** Nonsynonymous SNPs are separated into "Deleterious" vs. "Tolerated" and are plotted separately. The vertical grey line indicates the centromeric region.
(TIFF)

**S13 Fig. The number of nonsynonymous SNPs per covered codon in 10 Mb windows categorized as "common" in the landrace samples across the barley genome.** Nonsynonymous SNPs are separated into "Deleterious" vs. "Tolerated" and are plotted separately. The vertical grey line indicates the centromeric region.
(TIFF)

**S14 Fig. Representation of the CC and GG motif mutations that generate the most abundant amino acid changes identified as harmful in the SNVs.** Colors represent the polarity of each amino acid. Arrows show the within codon changes, their line thickness, and the side numbers indicate the correspondent Grantham score. A thicker line indicates a greater evolutionary distance between two amino acids. Red letters represent the mutations induced by the sodium azide-associated motifs CC or reverse complement GG. Colors represent the primary properties of each amino acid, including polarity and acidity.
(TIFF)

**S15 Fig. Frequency of amino acid changes for SNVs that annotate as tolerated versus deleterious in mutated lines.**
(TIFF)

**S16 Fig. A photograph of the mutagenized lines M28 next to M29, which has a distinct onion-like, compact, atypical barley phenotype.** Both lines have been self-fertilized for four generations, and photos were taken five weeks after planting.
(TIFF)

**S17 Fig. Grain yield for 25 mutated lines, the Morex W2017 parent, and eight check lines for three locations and three years.** The box plots are sorted by the median for each line, and the bars in the box plot indicate the mean. Red outlines represent sodium azide-treated lines. Red shaded boxes indicate mutated lines that were sequenced in this study.
(TIFF)

**S1 Table. Detailed summary of samples sequenced in this study, which samples were treated with sodium azide, the library preparation and sequencing technology, and mapped average coverage.**
(CSV)

**S2 Table. Log-linear analysis performed by Mutation Motif for C→T variants induced by sodium azide compared to those originating spontaneously in the reference sequence.** Position is relative to the mutated base. Deviance is a likelihood ratio from the log-linear model. Degrees-of-freedom (df) and P-values are from the chi-squared distribution.
(XLSX)

**S3 Table. The Pearson or Spearman correlation between the number of SNVs and the phenotypes across each functional class of variants.**
(CSV)

## Acknowledgments

The authors thank Ron Okagaki for help with the initial screening of the mutated lines; Lucie Lu for submitting the raw sequencing data to NCBI SRA; Elaine Lee for processing the 10X Genomics datasets using Longranger; Nadia Janis for scoring images of the deletions; Malik Samuel, Emily Vonderharr, Samuel Hamann, and Mackenzie Linane for help with growing out the first couple of generations and processing the seed; Erica Sun for making digital sketches of the barley plants, and Max Okagaki for help with S14 Fig. This research was carried out with software and hardware support provided by the Minnesota Supercomputing Institute (MSI) at the University of Minnesota. Syngenta Crop Protection Inc. provided DNA samples.

## Author contributions

**Conceptualization:** Chaochih Liu, Gary J. Muehlbauer, Kevin P. Smith, Justin C. Fay, Peter L. Morrell.

**Data curation:** Chaochih Liu.

**Formal analysis:** Chaochih Liu, Giulia Frascarelli.

**Funding acquisition:** Erik Legg, Gary J. Muehlbauer, Kevin P. Smith, Justin C. Fay, Peter L. Morrell.

**Investigation:** Chaochih Liu, Adrian O. Stec, Shane Heinen, Li Lei, Skylar R. Wyant, Kevin P. Smith.

**Methodology:** Chaochih Liu.

**Project administration:** Kevin P. Smith, Peter L. Morrell.

**Resources:** Chaochih Liu, Adrian O. Stec, Shane Heinen.

**Software:** Chaochih Liu, Giulia Frascarelli.

**Supervision:** Gary J. Muehlbauer, Kevin P. Smith, Peter L. Morrell.

**Validation:** Chaochih Liu, Giulia Frascarelli, Li Lei.

**Visualization:** Chaochih Liu.

**Writing – original draft:** Chaochih Liu, Peter L. Morrell.

**Writing – review & editing:** Chaochih Liu, Giulia Frascarelli, Li Lei, Monika Spiller, Gary J. Muehlbauer, Kevin P. Smith, Justin C. Fay, Peter L. Morrell.

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
