## [Decision Letter · Decision Letter 0]

Dear Dr Morrell,

Thank you very much for submitting your Research Article entitled 'Sodium azide mutagenesis induces a unique pattern of mutations' to PLOS Genetics.

<h4>The manuscript was fully evaluated at the editorial level and by independent peer reviewers. The reviewers appreciated the attention to an important problem, but raised some substantial concerns about the current manuscript. Based on the reviews, we will not be able to accept this version of the manuscript, but we would be willing to review a much-revised version. We cannot, of course, promise publication at that time.

If you decide to revise the manuscript for further consideration at PLOS Genetics, please aim to resubmit within the next 60 days, unless it will take extra time to address the concerns of the reviewers, in which case we would appreciate an expected resubmission date by email to plosgenetics@plos.org.

If present, accompanying reviewer attachments are included with this email; please notify the journal office if any appear to be missing. They will also be available for download from the link below. You can use this link to log into the system when you are ready to submit a revised version, having first consulted our Submission Checklist .

PLOS has incorporated Similarity Check , powered by iThenticate, into its journal-wide submission system in order to screen submitted content for originality before publication. Each PLOS journal undertakes screening on a proportion of submitted articles. You will be contacted if needed following the screening process.

To resubmit, log into your Editorial Manager account and select the option 'Revise Submission' in the 'Submissions Needing Revision' folder.

We are sorry that we cannot be more positive about your manuscript at this stage. Please do not hesitate to contact us if you have any concerns or questions.

Yours sincerely,

Angela Hancock, Ph.D.

Academic Editor

PLOS Genetics

Tanja Slotte

Section Editor

PLOS Genetics

**Associate Editor Comments:** </h4><h4>

Thank you for submitting your manuscript to PLoS Genetics, which has now been assessed by two expert reviewers. Comments from both reviewers were generally positive, but they identified some aspects that needed further attention. Many of these do not require additional analyses, but some minor additional analyses are also requested. Please address all comments from the two reviewers in your response.

Reviewer's Responses to Questions</h4>

**Comments to the Authors:**

Reviewer #1: The authors describe the analysis of 11 lines produced by Na azide mutagenesis. They distinguish induced mutations from the background providing spectrum, sequence context, and severity. The study appears fundamentally solid and is of interest because of the importance of induced mutations as a functional analysis and breeding tool, particularly with in the context of modern sequencing. Nevertheless, the manuscript requires several fixes, most in terms of clarity.

1. The manuscript language is often hard to follow, particularly in the results. The authors should make an effort to edit for clarity, stating the premises to each experiment or comparison, the expectations and the findings. I believe that each mutant line has 1000-4000 SNV, and 40 or 1500 ??? indels. Given the different methods for evaluating the indels, the results should specify what we can conclude from the analysis. For example, ~14 predicted severe mutants / line are observed. What about the genes affected by the indels? I think the information may be present, but it surely difficult to extract for a reader. I'm confused by the paragraph in Discussion ending in "This is a ~130-140 fold increase in the index mutation rate to an average of 5.13 x 10-7." What mutation rate? Similarly, I would like the results to make it clear where the following conclusion is coming from: "The typical line has an average of 273.3 disruptions of coding variants, including SNVs and indels."

2. Figure 1 needs better explanation. For example, the untreated rare category seems to be referring to novel natural variants. However, I am not clear why some would occur in two lines and why there are so many of them. How many are estimated per generation? It seems that the lines are separated by ~ 5 generations (less than 10 in any case). How can 2M occur in such a short time if the previous standing variants (different from reference) are 4M? Further, all those that occurred in the last generation should be heterozygous (half of those in the previous, 1/4 of the one before, etc etc) and should provide an estimate of the per generation natural mutation.

3. Compared to EMS, this a relative low frequency of lesions. What I could not figure out, is how many mutations were observed to accumulate spontaneously per generation: 124 are predicted per line. Given the C->T predominance in the induced mutations compared to natural, it is clear that the authors are distinguishing the two categories. One criterion is that mutations are private (i.e. affect a single individual) while most natural variation is preexistent and common. New natural mutations, of course, would be private. Could the numbers be provided per line to make this clear? Could the criteria for distinguishing induced from natural changes be made clearer?

4. Heterozygous for homozygous mutations. The selfing stage of the tested material predicts that most mutations and even more of the preexistent variation should be homozygous. The authors should present data addressing zygosity. I am not clear what filter was used to call a mutation. In other words, what minimum coverage was used for homozygous and heterozygous calls. The sequencing coverage has a large effect (Henry, 2014).

5. Nomenclature of generations after mutagenesis. The authors seem to call the first generation produced directly from treated seed the M0 (See introduction: "Mutagenized plants are identified as the M0. "). On the other hand, Figure S1 could be interpreted to use M1 for the treated seed. Which applies? If the authors use M0, this is not a standard usage in barley, wheat, rice or arabidopsis and will generate confusion. This generation is commonly called the M1. See for example Gottwald, Barley, 2009; Hasegawa, Barley, 1984; Kleinhofs, Barley, 1978; Perry, Lotus, 2009; Green, Arabidopsis, 2003; Henry, Wheat, rice, 2014.

6. The statement that the two round of selfing are needed to perform phenotypic screening is not supported by genetic theory or by the literature. The authors cite an FAO manual, which I could not access easily, but I doubt that it provides evidence toward this claim. Regardless, most screens are performed at the canonical M2 generation (one round of selfing), which already ensures that the mutations are meiotically inherited. One of course may elect to use the M3, but it is a matter of preference.

7. The issue of what LD50 is optimal is not easily determined. In many systems, chromosomal aberration contribute to the LD50 and the extent of genomic damage does not necessarily predict the point mutation density. The authors, in fact, provide direct evidence toward this because Na azide mutagenesis tends to induce fewer stop codons and splice site mutations than EMS. Therefore, the rate of lethal KO from mutagenesis is expected to be different for these two systems.

8. The statement "This suggests that there are roughly 14 phenotype-changing variants per individual treated line." Should be rephrased because it is unlikely to be true for the following reasons: the majority of KO's in a diploid organism do not have a readily detected phenotype (see for example the experience with the homozygous mutant collection at Salk). Predicted deleterious mutations among missense ones have an even lower probability of phenotypic consequences. I think the author meant that 14 phenotypic changes are theoretically possible and consistent with the yield reduction observed. Indeed, only one mutant has a readily visible phenotype (i.e. not yield related).

Reviewer #2: the review is uploaded as an attachment

**Have all data underlying the figures and results presented in the manuscript been provided?**

Reviewer #1: Yes

Reviewer #2: None

PLOS authors have the option to publish the peer review history of their article (what does this mean? ). If published, this will include your full peer review and any attached files.

**Do you want your identity to be public for this peer review?** For information about this choice, including consent withdrawal, please see our Privacy Policy .

Reviewer #1: No

Reviewer #2: No

---

## [Decision Letter · Decision Letter 1]

Dear Dr Morrell,

We are pleased to inform you that your manuscript entitled "Sodium azide mutagenesis induces a unique pattern of mutations" has been editorially accepted for publication in PLOS Genetics. Congratulations!

Yours sincerely,

Angela Hancock, Ph.D.

Section Editor

PLOS Genetics

Tanja Slotte

Section Editor

PLOS Genetics

Aimée Dudley

Editor-in-Chief

PLOS Genetics

Anne Goriely

Editor-in-Chief

PLOS Genetics

Comments from the reviewers (if applicable):

Both reviewers are satisfied with the current version and responses. I recommend the manuscript be accepted.

Reviewer's Responses to Questions

**Comments to the Authors:**

Reviewer #1: I am satisfied by the authors' answers and revision. I thank the authors for the careful work.

Reviewer #2: I think the answers to my questions are acceptable. I agree with the authors for the reply of my comments.

**Have all data underlying the figures and results presented in the manuscript been provided?**

Reviewer #1: Yes

Reviewer #2: Yes

PLOS authors have the option to publish the peer review history of their article (what does this mean? ). If published, this will include your full peer review and any attached files.

**Do you want your identity to be public for this peer review?** For information about this choice, including consent withdrawal, please see our Privacy Policy .

Reviewer #1: No

Reviewer #2: **Yes: ** xiaorng Fan

**Data Deposition**

http://datadryad.org/submit?journalID=pgenetics&manu=PGENETICS-D-24-00537R1

**Press Queries**

---

## [Editor Report · Acceptance letter]

PGENETICS-D-24-00537R1

Sodium azide mutagenesis induces a unique pattern of mutations

Dear Dr Morrell,

We are pleased to inform you that your manuscript entitled "Sodium azide mutagenesis induces a unique pattern of mutations" has been formally accepted for publication in PLOS Genetics! Your manuscript is now with our production department and you will be notified of the publication date in due course.

With kind regards,

Anita Estes

PLOS Genetics

On behalf of:
